# Research

computational chemistry/computer modelling and simulation/atomic and molecular physics

bimetallic clusters, stable structures, geometrical optimization, surface distribution, core/shell configuration

**Author for correspondence:**
Xia Wu
e-mail: xiawu@aqnu.edu.cn

This article has been edited by the Royal Society of Chemistry, including the commissioning, peer review process and editorial aspects up to the point of acceptance.

# Theoretical study of the structures of bimetallic Ag–Au and Cu–Au clusters up to 108 atoms

Rongbin Du, Sai Tang, Xia Wu, Yiqing Xu, Run Chen and Tao Liu

AnHui Province Key Laboratory of Optoelectronic and Magnetism Functional Materials, Key Laboratory of Functional Coordination Compounds of Anhui Higher Education Institutes, School of Chemistry and Chemical Engineering, Anqing Normal University, Anqing 246011, People's Republic of China

XW, 0000-0002-1143-6024

The stable structures of Ag–Au and Cu–Au clusters with $1:1$, $1:3$ and $3:1$ compositions with up to 108 atoms are obtained using a modified adaptive immune optimization algorithm with Gupta potential. The dominant motifs of Ag–Au and Cu–Au clusters are decahedron and icosahedron, respectively. However, in Ag-rich Ag–Au clusters, more icosahedra are found, and in Cu-rich Cu–Au clusters, there exist several decahedral motifs. Four Leary tetrahedral motifs are predicted. $Cu_{core}Au_{shell}$ configurations are predicted in Cu–Au clusters. In Ag–Au clusters, most Ag atoms are on the surface, but partial ones are located in the inner shell, while Au atoms are interconnected in the middle shell.

## 1. Introduction

Nanoclusters have been extensively studied by experimental and theoretical studies because of their great importance in medicine, biology, catalysis, optics and electronics [1–3]. In the field of nanomaterials, multi-metallic clusters and their compounds have increased interest for wider range of properties by mixing two or more chemical elements, which often exhibit enhanced catalytic reaction performance compared to those of the pure metals [4–7]. For instance, for the reduction of 4-nitrophenol by $NaBH_4$, the catalytic activity of Ag–Au nanoparticles was higher than Ag and Au monometallic ones [4]. The structural morphology, chemical ordering or composition have a substantial influence on the properties of multi-metallic clusters, e.g. kinetic stability [8], the activity and selectivity of a reaction [1].

For Ag–Au and Cu–Au clusters, exhaustive experimental and theoretical studies have been carried out to reveal structural

details [9–16]. It was found that the nanoparticle architecture depended on the surfactants [9], and Ag–Au nanoparticles with core/shell and alloy configurations could be formed while the surfactants are hexadecyltrimethylammonium chloride and sodium dodecyl sulfate, respectively. The synthesized $Au_{core}Ag_{shell}$ nanoparticles could be tuned by varying sizes and compositions independently [10,11]. However, $Ag_{core}Au_{shell}$ nanoparticles could be an excellent alternative for surface-enhanced Raman scattering measurements [12,13]. In a theoretical study, the results by density functional theory (DFT) showed that the doping of Au atoms improved the stability of Ag–Au clusters [17]. A combination of global optimization, i.e. Birmingham cluster genetic algorithm (BCGA), and DFT calculations was applied to study Ag–Au clusters [18–20]. Results showed that the transition from two- to three-dimensional structures was predicted between $Au_6Ag_2$ and $Au_5Ag_3$ [18], and the atomic ordering in core/shell structures was found to be related to the electric dipole moment [19]. There existed apparent tendency for surface segregation of the Ag atoms in 38-atom Ag–Au clusters, and the stability of clusters was related to the increasing number of Au–Au and Ag–Au bonds [20]. In the stable structures of Ag–Au clusters containing 20–150 atoms (with atomic ratio 1 : 1), decahedra (Dh) and icosahedra (Ih) were the main motifs [21], and $Ag_{44}Au_{44}$ cluster was deemed to have high structural, electronic and thermal stability. Furthermore, a cluster expansion model was used to determine the chemical ordering of 309-atom Ag–Au Mackay icosahedral nanoparticles [22].

In the theoretical study of Cu–Au clusters, the structural, energetic and electronic properties of the $Cu_nAu_{26-n}$ clusters have been carried out by DFT calculations [23]. A molecular dynamic (MD) study of the 256-atom Cu–Au clusters using the Gupta potential found that Au doping of Cu clusters led to a clear decrease of the surface energy [24]. In a study of the coefficient of thermal conductivity of 55-atom Cu–Au clusters by MD simulations with the quantum Sutton–Chen (SC) potential, it was found that melting temperature of Cu–Au clusters increased with Cu atom fraction [25]. The distribution with a Cu-rich core/Au-rich surface was found in stable icosahedral structures at sizes 4 and 10 nm by thermodynamic approach with the quantum SC potential [26], as well as with energy calculations of the structures of large-scale Cu–Au clusters up to 561 atoms modelled by the Gupta potential [27]. On the other hand, based on the Gupta potential, the stable structures of Cu–Au clusters with up to 56 atoms were investigated using BCGA [28], and the structures of Cu–Au clusters around $CuAu_3$, CuAu and $Cu_3Au$ compositions were also studied [29].

Previously, we have studied the stable geometrical structures of 55-atom Cu–Au (i.e. 37 icosahedra and 18 amorphous structures) and Ag–Au clusters (i.e. 55 Mackay icosahedra) by using a global optimization algorithm, i.e. a modified adaptive immune optimization algorithm (AIOA), on the basis of the many-body Gupta potential [30]. It was found that the stability of the bimetallic clusters was affected by different size, chemical composition and symmetry. To investigate the influence of Ag and Cu metallic dopants on the stable structures of Au-based clusters, we perform the comparison on the most stable structures of Ag–Au and Cu–Au clusters up to 108 atoms. The stable structures of those clusters with atomic ratio 1 : 1, 1 : 3 and 3 : 1 are located by using a modification algorithm of AIOA (called AIOA-IC algorithm), i.e. AIOA based on the construction of inner cores, and the Gupta potential. Their structural characteristic and atomic distribution in M–Au (M = Cu and Ag) clusters are studied. Furthermore, the influence of different ratio of Ag and Cu atoms on the motifs of Au-based clusters is analysed.

# 2. Method

## 2.1. Gupta potential for Ag–Au and Cu–Au clusters

The Gupta potential, as formulated by Cleri & Rosato [31], is adopted to describe the interatomic interactions in bimetallic Ag–Au and Cu–Au clusters. It is derived from a second-moment approximation to a tight-binding Hamiltonian. The Gupta potential ($V_N$) with $N$ atoms has the following form:

$$V_N = \frac{1}{2} \sum_{i=1}^{N} \{V^r(i) - V^m(i)\}, \tag{2.1}$$

$$V^r(i) = \sum_{j=1(j \neq i)}^{N} A_{ij} \exp\left(-p_{ij}\left(\frac{r_{ij}}{r_{ij}^{(0)}} - 1\right)\right) \tag{2.2}$$

and

$$V^m(i) = \left[\sum_{j=1(j \neq i)}^{N} \xi_{ij}^2 \exp\left(-2q_{ij}\left(\frac{r_{ij}}{r_{ij}^{(0)}} - 1\right)\right)\right]^{1/2}, \tag{2.3}$$

**Table 1.** The Gupta potential parameters for bimetallic Ag–Au and Cu–Au clusters.

| parameters | Ag–Au clusters | | | Cu–Au clusters | | |
| --- | --- | --- | --- | --- | --- | --- |
| | Ag–Ag | Ag–Au | Au–Au | Cu–Cu | Cu–Au | Au–Au |
| $A_{ij}$ (eV) | 0.1031 | 0.1488 | 0.2096 | 0.0855 | 0.1539 | 0.2061 |
| $\xi_{ij}$ (eV) | 1.1895 | 1.4874 | 1.8153 | 1.224 | 1.5605 | 1.79 |
| $p_{ij}$ | 10.85 | 10.494 | 10.139 | 10.96 | 11.05 | 10.229 |
| $q_{ij}$ | 3.18 | 3.607 | 4.033 | 2.278 | 3.0475 | 4.036 |
| $r_{ij}^{(0)}$ (Å) | 2.8921 | 2.8885 | 2.885 | 2.556 | 2.556 | 2.884 |

where $r_{ij}$ represents the distance between atoms of species $i$ and $j$. $V^r(i)$ is a pairwise repulsive term, and $V^m(i)$ is the attractive many-body terms. In this study, all the parameters for Ag–Au and Cu–Au clusters are taken from [31,32] as listed in table 1.

## 2.2. Optimization algorithm

The AIOA method is a global search technique based on the evolutionary ideas of clonal selection principles and biological immune systems [33]. In the domain of structural optimization, it has been successfully applied for locating the stable structures of monoatomic clusters, e.g. Lennard–Jones (LJ)$_{3-200}$ clusters [34], binary clusters, e.g. Cu–Au [30] and Ag–Pd [35] clusters, ternary clusters such as Ar–Kr–Xe [36] and Au–Pd–Pt clusters [37], and quaternary Ag–Au–Pd–Pt clusters [38]. The basic step of AIOA includes generating initial structures, clone selection, mutation operation and updating operation. At first, a certain number ($n_{lib}$) of initial configurations are randomly generated and locally minimized [39], forming the original gene library. Then by an immune clone selection procedure, a population ($n_{pop}$) of individuals is selected from the gene library. In binary clusters, there exist geometrical isomers and homotopic isomers, i.e. the same configurations with different atomic type arrangement. In the mutation operation, to solve geometrical isomers problem, half of these individuals are carried out with the energy-based mutation, and for the other half, two types of atoms are randomly selected and their locations are exchanged to solve the homotopic problem. Energy-based mutation is designed based on the fact that the atoms with lower number of nearest-neighbour contacts generally have higher potential energies. For each atom, the probability to be mutated is with respect to the number of nearest-neighbour contacts. Then, the selected atom is moved to a random site on the surface of the cluster. New individuals are thus generated. Next, a similarity checking method is designed to update the gene library, in which the new individuals with less similarity and lower energy are kept by using connectivity table (CT) [34]. At last, the cycle of the clone selection, mutation operation and updating operation repeats $n_{rep}$ times to find the global minima.

A strategy by fixing the inner cores of the starting structures has played an important role in determining the structures of large-scale monoatomic clusters, and LJ clusters up to 150 atoms by Hartke [40]; LJ$_{670}$ [41] and Al$_{510}$ [42] clusters are thus optimized. The idea is also adopted in the AIOA-IC method, which is a modification of AIOA. Besides randomly generating the atomic coordinates as in AIOA, an inner core is also constructed while building the starting structure in AIOA-IC. Furthermore, decahedron (Dh), icosahedron (Ih), face centred cubic (fcc), sixfold pancake and Leary tetrahedron (LT) are the main motifs in atomic clusters as plotted in figure 1, and they are selected as their inner cores as introduced in [43]. Then around the inner core, the remaining atoms are randomly dispersed to form starting structures. Therefore, $n_{lib}$ initial configurations are randomly selected from random generation of atomic coordinates and different core structures. The developed AIOA-IC method has been applied for locating the stable structures of trimetallic Cu–Au–Pt clusters [43] with 147 atoms. In this work, AIOA-IC is used for M–Au (M = Ag and Cu) clusters with $N = 60$–108, and 55-atom Dh, 55-atom Ih, 44- and 88-atom fcc, 51-atom sixfold pancake-like, 34- and 98-atom LT cores are adopted. Moreover, the following parameters are employed: $n_{lib} = 20$, $n_{pop} = 15$, and $n_{rep} = 1500$. On the other hand, the procedure of AIOA-IC should run $n_{run} = 100$ times.

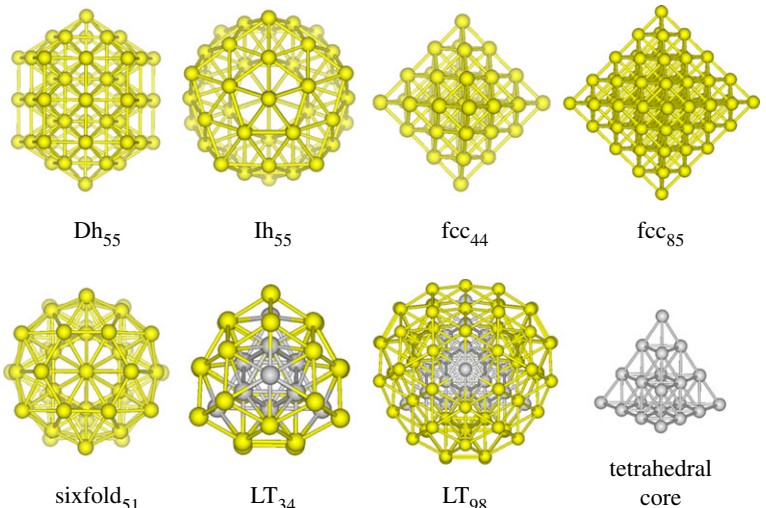

**Figure 1.** Typical motifs of decahedral (Dh), icosahedral (Ih), face centred cubic (fcc), sixfold and Leary tetrahedron (LT). Tetrahedral core of LT$_{98}$ is also shown. The inner cores of LT$_{34}$ and LT$_{98}$ configurations are represented by grey spheres.

**Table 2.** Potential energies for Ag$_n$Au$_n$ ($n$ = 30–54) clusters. Lower energies are in bold.

| $N$ | $E$ (eV) | $E_{ref}$ (eV)[a] | $n$ | $E$ (eV) | $E_{ref}$ (eV)[a] |
|---|---|---|---|---|---|
| 30 | **−188.3738** | −188.1643 | 43 | **−272.8780** | −272.3030 |
| 31 | **−194.8413** | −194.7674 | 44 | **−279.6674** | −279.1388 |
| 32 | **−201.5975** | −201.1421 | 45 | **−286.2538** | −285.5936 |
| 33 | **−207.9968** | −207.7146 | 46 | **−292.7457** | −292.0234 |
| 34 | **−214.4447** | −214.0908 | 47 | **−299.3887** | −298.8105 |
| 35 | **−221.0684** | −220.7654 | 48 | **−306.1635** | −305.3793 |
| 36 | **−227.6291** | −227.1792 | 49 | **−312.7878** | −312.1650 |
| 37 | **−234.1900** | −233.8269 | 50 | **−319.1323** | −318.7294 |
| 38 | **−240.7105** | −240.3646 | 51 | **−325.6465** | −325.4572 |
| 39 | **−247.0362** | −246.6945 | 52 | **−332.5967** | −332.0059 |
| 40 | **−253.4077** | −253.0306 | 53 | **−338.8598** | −338.4055 |
| 41 | **−259.9580** | −259.4811 | 54 | **−345.7535** | −344.7734 |
| 42 | **−266.5060** | −265.7221 | | | |

[a]The values of $E_{ref}$ are taken from [21].

# 3. Results and discussion

## 3.1. Stable structures of Ag$_n$Au$_n$ and Cu$_n$Au$_n$ clusters

The putative stable structures of M–Au clusters are optimized by using AIOA-IC method. Previously, the stable structures of Ag–Au clusters of 1 : 1 composition up to 150 atoms were determined by BCGA [21] using the same Gupta potential and parameters. In order to confirm that the same parameters are used, Ag$_n$Au$_n$ ($n$ = 10–30) clusters are optimized, and the same structures and energies are obtained. Furthermore, the stable structures of Ag$_n$Au$_n$ ($n$ = 30–54) clusters are reproduced, and new lower energy minima are found in this work. The potential energies of the investigated Ag–Au clusters are listed in table 2, and as a comparison the corresponding potential energies reported in [21] are also listed in the table. In the table, the lower energies are labelled in bold font, and it is clear that all clusters found in this study have lower energies than those reported. The maximum difference between energy values is about 0.9801 eV at Ag$_{54}$Au$_{54}$, and the minimum difference (about 0.0739 eV) appears at Ag$_{31}$Au$_{31}$. Such results are also a proof for the high efficiency of the AIOA-IC method for

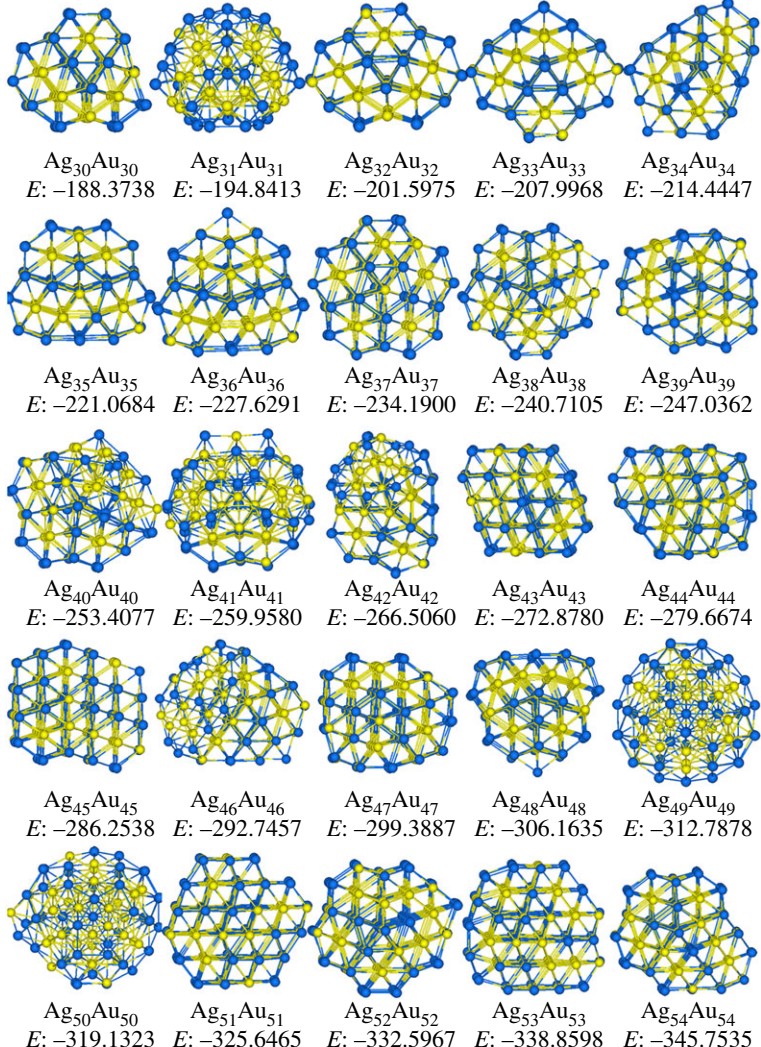

**Figure 2.** Stable structures of $Ag_nAu_n$ ($n = 30$–$54$) clusters, and Ag and Au atoms are represented by blue and yellow spheres, respectively, and the potential energies $E$ (eV) are also added.

bimetallic clusters optimization. Nevertheless, it should be noted that the cut-off constraint in the Gupta potential reflects the pairwise repulsive term and the attractive many-body terms, i.e. potential energies [44]. In our studies, the Gupta potential does not include cut-off constraint, and the relatively higher energies obtained in [21] may be due to the cut-off values.

Figures 2 and 3 plot the putative stable structures of $Ag_nAu_n$ and $Cu_nAu_n$ ($n = 30$–$54$) clusters. In figure 2, structures of $Ag_nAu_n$ clusters can be categorized into 18 Dh at $n = 30$, 32–40, 42, 44, 46–48, 52–54, one distorted Ih at $Ag_{31}Au_{31}$, one twinned Dh at $n = 41$, three stacking-fault fcc (sf-fcc) at $n = 43$, 45 and 51, and two LT at $Ag_{49}Au_{49}$ and $Ag_{50}Au_{50}$. LT motifs were discovered as the global minima of $LJ_{98}$ [45], $Ni_{98}$ [46] and partial 98-atom Pd–Pt clusters [47], exhibiting $T_d$ point group. Apparently, the dominant motif of the studied $Ag_nAu_n$ clusters is Dh. Compared with the results in previous work [21], it is found that there exist significant differences in structures. For instance, both $Ag_{49}Au_{49}$ and $Ag_{50}Au_{50}$ clusters have LT motifs in this study, but they are recognized as Dh motifs in the literature. It can also be seen in figure 2 that most of Ag atoms are distributed on the surface of the structures. In figure 3, it is clear that all $Cu_nAu_n$ clusters take the icosahedral motifs based on 55-atom Mackay icosahedron. The segregation of Au atoms to the surface in Cu–Au clusters can be found in figure 3.

It should be noted that although the random generation of atomic coordinates is also retained in the AIOA-IC algorithm, the algorithm is biased to a certain extent. It is shown that when the optimal configuration of the cluster with the lowest energy is the same as one of the initial core configurations, it is easier to search the configuration from the corresponding initial core. For example, the optimal structure with Ih motif is mostly derived from the Ih cores. However, the optimization process shows that Ih motifs can also be obtained from the Dh cores because of the structural transformation from Dh to Ih [32].

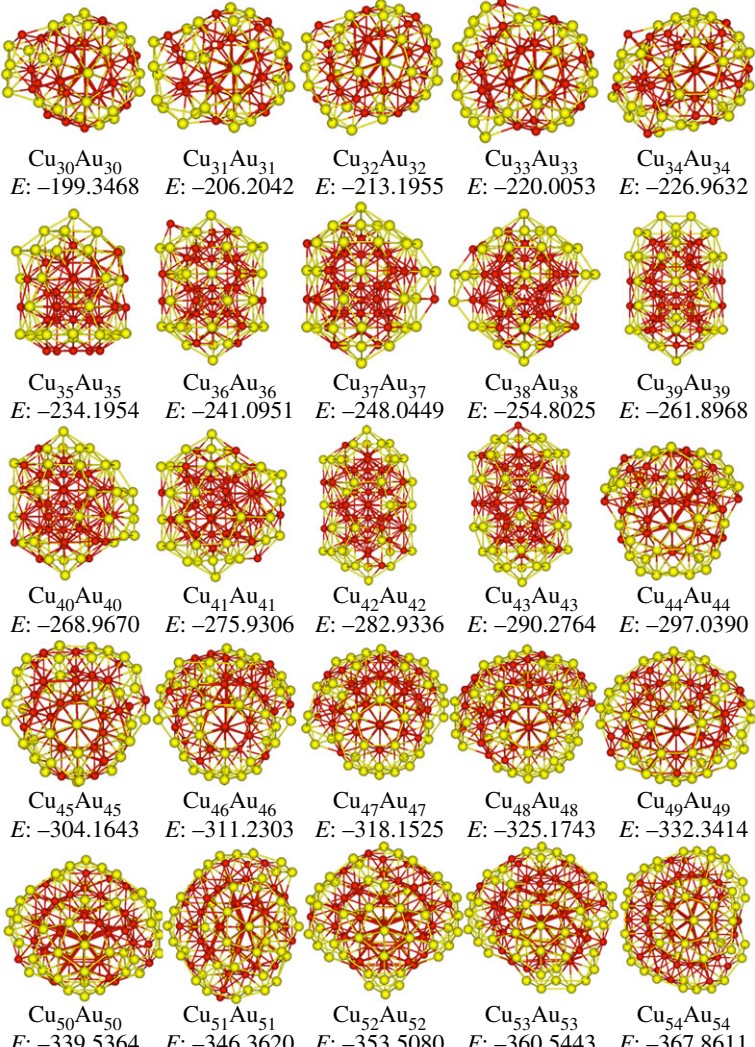

**Figure 3.** Stable structures of $Cu_nAu_n$ ($n = 30$–54) clusters, and Cu and Au atoms are represented by red and yellow spheres, respectively, and the potential energies $E$ (eV) are also added.

## 3.2. Stable structures of $Ag_nAu_{3n}$ and $Cu_nAu_{3n}$ clusters

Figure 4 shows the stable structures of $Ag_nAu_{3n}$ and $Cu_nAu_{3n}$ ($n = 15$–27) clusters. In $Ag_nAu_{3n}$ clusters, structures include eight Dh at $n = 15$–19, 23, 24 and 27, three Dh with anti-layers at $n = 20$, 25 and 26, one fcc at $Ag_{22}Au_{66}$, and one sf-fcc at $Ag_{21}Au_{63}$. Therefore, the main structure of $Ag_nAu_{3n}$ clusters is Dh as discussed in $Ag_nAu_n$ clusters plotted in figure 2. On the other hand, at $Cu_{15}Au_{45}$ in figure 4, ring-like structure linked by three face-sharing double icosahedra [48] is formed. In the size range of $n = 16$–18, clusters take the icosahedral motifs. At $Cu_{19}Au_{57}$, four face-sharing double icosahedra are linked to form a ring-like structure. With the increase of $n = 20$–27, all clusters have icosahedral configurations. Therefore, icosahedral is still the main motif in the investigated $Cu_nAu_{3n}$ clusters as found in $Cu_nAu_n$ clusters.

## 3.3. Stable structures of $Ag_{3n}Au_n$ and $Cu_{3n}Au_n$ clusters

Figure 5 shows the stable structures of $Ag_{3n}Au_n$ and $Cu_{3n}Au_n$ ($n = 15$–27) clusters. In $Ag_{3n}Au_n$ clusters, at $Ag_{45}Au_{15}$, an icosahedral motif is formed. In the size range of $n = 16$–20, clusters have decahedral configurations. With the increase of $n$, at $Ag_{63}Au_{21}$, $Ag_{66}Au_{22}$ and $Ag_{72}Au_{24}$, partial icosahedra based on 147-atom Mackay icosahedron are formed. The motifs of $Ag_{69}Au_{23}$ and $Ag_{75}Au_{25}$ clusters are Dh, which is recognized as (3,3,2)-Dh [49]. The form of $(m,n,l)$-Dh is used to define Marks' decahedron, where parameters $m$ and $n$ denote the width and height of the rectangular (100) faces, and $l$ represents the depth of the Marks re-entrance. At $Ag_{78}Au_{26}$, the structure is grown based on 98-atom

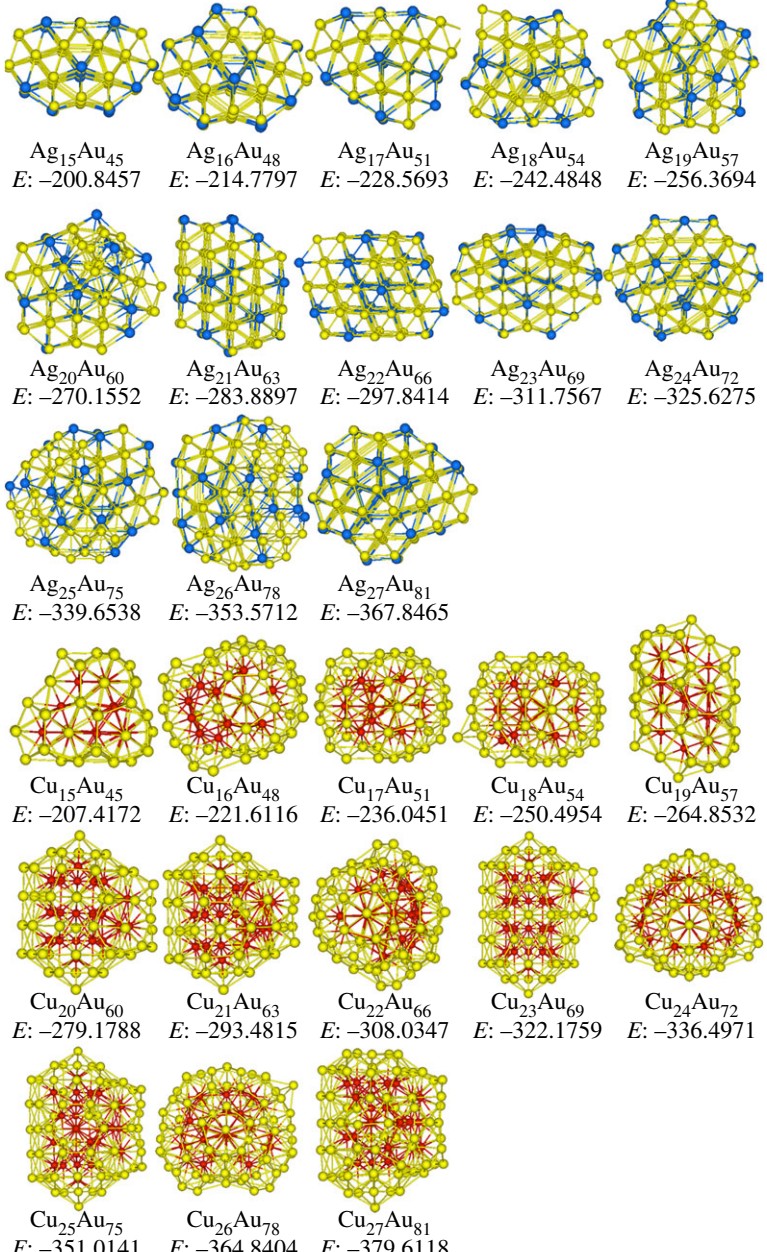

**Figure 4.** Stable structures of Ag$_n$Au$_{3n}$ and Cu$_n$Au$_{3n}$ ($n = 15$–27) clusters, and Cu, Ag and Au atoms are represented by red, blue and yellow spheres, respectively, and the potential energies $E$ (eV) are also added.

LT configuration. At last, at Ag$_{81}$Au$_{27}$ cluster, a Dh motif with anti-layers is formed. Therefore, it can be seen that in Ag-rich regions of Ag–Au clusters, more icosahedral structures are found than in Au-rich regions as discussed in Ag$_n$Au$_{3n}$ clusters.

In Cu$_{3n}$Au$_n$ clusters as plotted in figure 5, structures can be categorized into eight Ih at $n = 15$, 16, 20–24 and 27, four Dh at $n = 17$–19 and 26, and one LT at Cu$_{75}$Au$_{25}$. It is apparent that Ih is still the main motif as in the studied Cu$_n$Au$_{3n}$ clusters. However, in Cu-rich contents of Cu–Au clusters, more Dh structures are found than in Au-rich contents as discussed in Cu$_n$Au$_{3n}$ clusters (demonstrated in figure 4). Furthermore, around 98 atoms, e.g. Cu$_{75}$Au$_{25}$ cluster, an LT structure is found, which is not found in Cu$_n$Au$_{3n}$ and Cu$_n$Au$_n$ clusters.

On the other hand, it should be noted that over the 100 independent runs of the AIOA-IC method, all the stable structures of the investigated Ag–Au and Cu–Au clusters are located with the successful rate above 2/100. The successful rates for some clusters are as high as 20%. It provides a proof for the efficiency of AIOA-IC method for the structural optimization of Ag–Au and Cu–Au clusters.

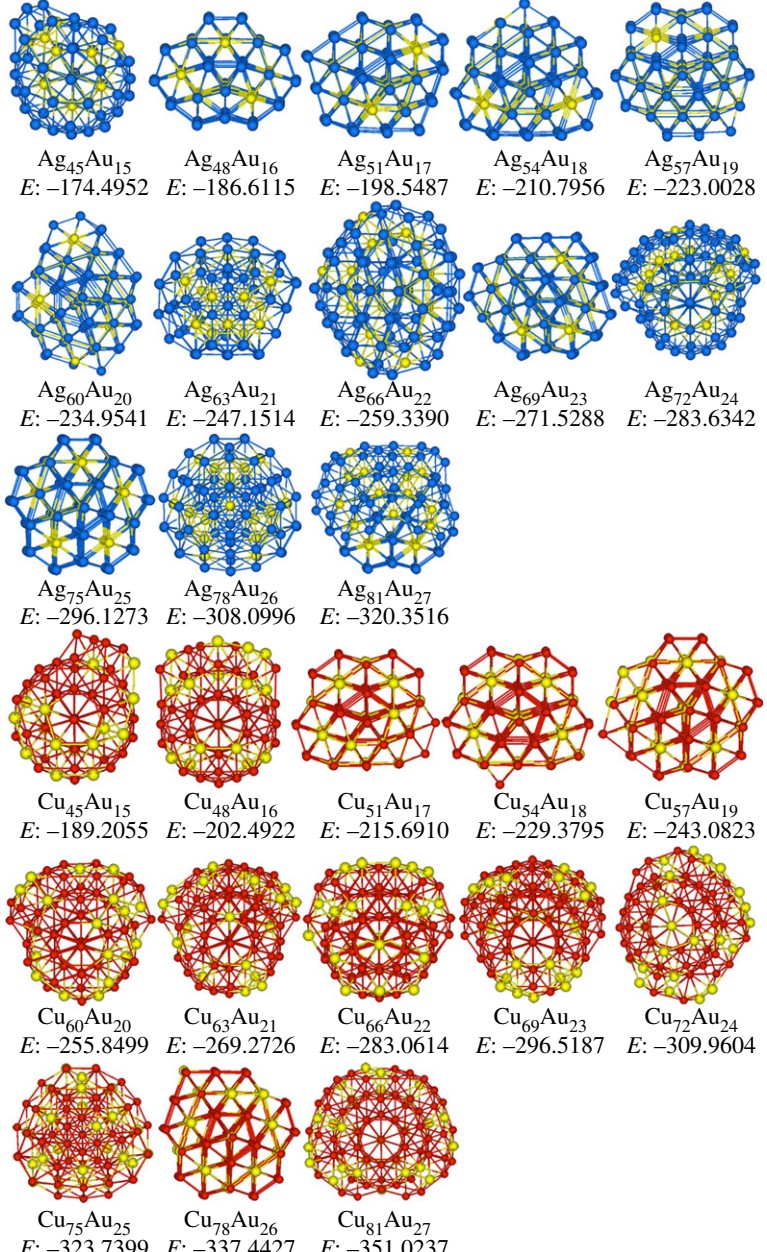

**Figure 5.** Stable structures of Ag$_{3n}$Au$_n$ and Cu$_{3n}$Au$_n$ ($n = 15$–27) clusters, and Cu, Ag and Au atoms are represented by red, blue and yellow spheres, respectively, and the potential energies $E$ (eV) are also added.

## 3.4. Analysis of atomic distribution

The order parameter ($R$) in binary A–B clusters is adopted to explain the atomic distribution or mixing degree of different elements. Actually, $R$-value is measured by the average distance of a type of atom (A or B) from the centre of a cluster, i.e.

$$R_A = \frac{1}{n_A} \sum_{i=1}^{n_A} \sqrt{x_i^2 + y_i^2 + z_i^2} \tag{3.1}$$

where $n_A$ denotes the number of the atoms of type A in the binary A–B clusters, and $x_i$, $y_i$ and $z_i$ represent their atomic coordinates. Generally, a small or large $R$-value means that the corresponding type atoms are at the centre or surface of the cluster having segregated pattern, respectively, and a medium value explains a mixed form.

Figure 6 shows the variation of the order parameter $R$ and its standard deviation (s.d.) values of Ag and Au atoms in Ag$_n$Au$_n$ clusters (figure 6a) and Cu and Au atoms in Cu$_n$Au$_n$ clusters (figure 6b) along with the $n$-value. From the curve of figure 6a, $R_{Ag}$ is slightly larger than $R_{Au}$ in Ag$_n$Au$_n$ clusters. It means

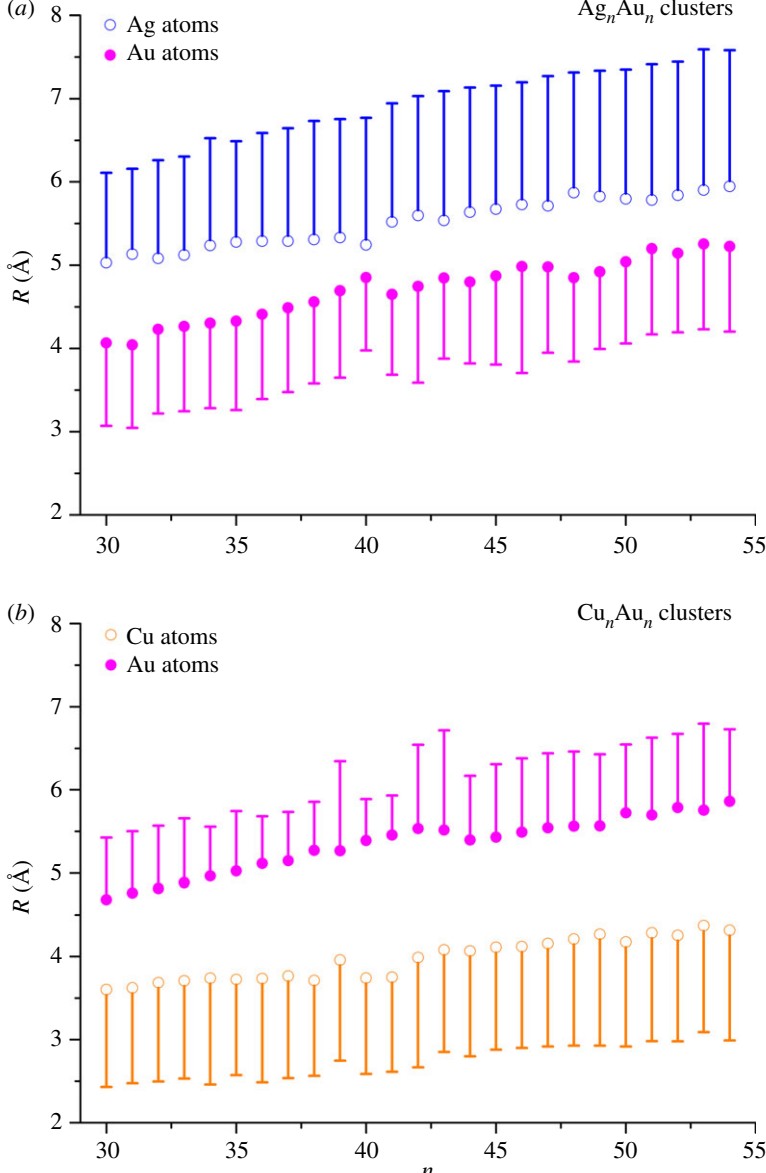

**Figure 6.** Average distance ($R$) of Ag and Au atoms from the centre of $Ag_nAu_n$ cluster (a) and Cu and Au atoms from the centre of $Cu_nAu_n$ cluster (b) with the standard deviation. Error bars show the standard deviation. For clarity, only half the error bar is shown.

that compared with Au atoms, Ag atoms are more inclined to be in the outer shell. Relatively large s.d. values for $R_{Ag}$ indicate that partial Ag atoms are also located in the inner shell. For Au atoms, their $R_{Au}$ values are medium, and the corresponding s.d. is small. It indicates that Au atoms are mainly distributed in the middle shell in a more compact way. In figure 6b of $Cu_nAu_n$ clusters, $R_{Au}$ values are clearly larger than those of $R_{Cu}$. Therefore, for Cu–Au clusters there exist significant surface segregation with Au atoms on the surface and Cu atoms in the core. On the other hand, the difference between $R_{Au}$ and $R_{Cu}$ in 1:1 Cu–Au clusters is bigger than that between $R_{Ag}$ and $R_{Au}$ in 1:1 Ag–Au clusters.

The number of bonds, i.e. the nearest-neighbour contacts ($n_{ij}$), can be further calculated to analyse the atomic distribution between homogeneous atoms or heterogeneous atoms. The calculation of $n_{ij}$ is given by

$$n_{ij} = \sum_{i<j} \delta_{ij},$$
(3.2)

where $\delta_{ij} = \begin{cases} 1, & r_{ij} \leq 1.2r_{ij}^{(0)} \\ 0, & r_{ij} > 1.2r_{ij}^{(0)} \end{cases}$ $i, j =$ Ag and Au in Ag–Au clusters, or Cu and Au in Cu–Au clusters,

and $r_{ij}^{(0)}$ is a nearest-neighbour criterion described above.

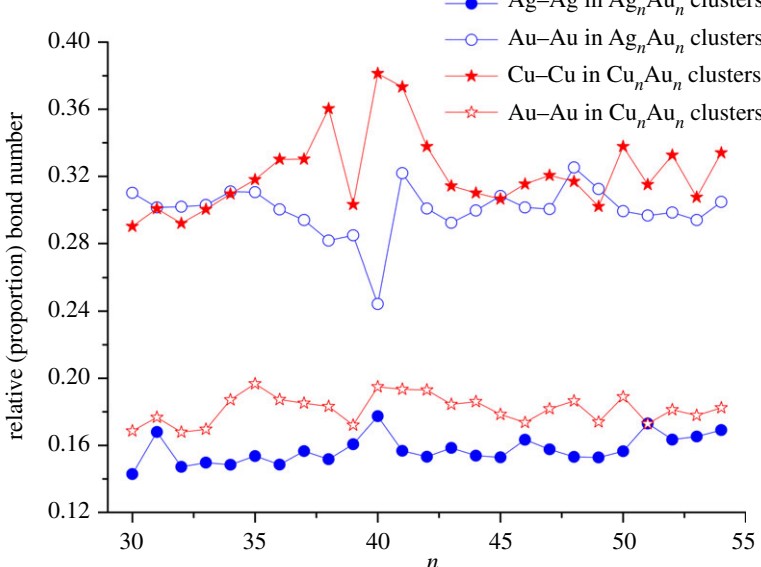

**Figure 7.** The proportion of bond number of Ag–Ag and Au–Au bonds to the total number of bonds in $Ag_nAu_n$ ($n = 30$–54) clusters and the corresponding ones of Cu–Cu and Au–Au in $Cu_nAu_n$ clusters.

Relative bond number, i.e. the proportion of Ag–Ag (or Cu–Cu) and Au–Au bond number to total bond number in $Ag_nAu_n$ (or $Cu_nAu_n$) clusters is plotted in figure 7. In the figure, the relative number of Au–Au bond is significantly larger than that of Ag–Ag bond in $Ag_nAu_n$ clusters. It is because Ag atoms tend to be on the surface, and Au atoms are interconnected with each other in the middle shell. The conclusion is consistent with the analysis by $R_{Ag}$ and $R_{Au}$ and their s.d. values above. Furthermore, from the figure, the relative number of Cu–Cu bond is significantly larger than that of Au–Au bond in $Cu_nAu_n$ clusters. It can be explained by the fact that Cu and Au atoms are located in the inner shell and outer shell, respectively. The segregation phenomena of Cu and Au in Cu–Au clusters can be explained in terms of larger surface energy of Cu (114 meVÅ$^{-2}$) compared to Au (96.8 meVÅ$^{-2}$) [37]. The surface energy of Ag (78 meVÅ$^{-2}$) [50] is smaller than that of Au, so most Ag atoms tend to occupy the surface sites.

## 3.5. Comparison with previous studies

The surface segregation phenomenon of Ag atoms in Ag–Au clusters has been predicted using genetic algorithm with the Gupta potential [20], which is in agreement with our simulated results. In a study of 1 : 1 Ag–Au clusters, in the size range 20–66, the structures were in favour of icosahedral motifs, and in the size range 68–128, they changed to be decahedra [21]. The tendency of forming decahedron is consistent with our modelling results in this work. However, it should be noted that previous studies showed that the Gupta model with the present parameters was insufficiently accurate to predict the degree of segregation or mixing for Ag–Au clusters, because the charge transfer effects were not considered [19,51–53]. In a reparametrization of Gupta model developed by taking into account such effects, larger proportion of Au atoms on the surface was observed than in the present work [54]. In addition, recent experiments for Ag–Au nanoparticles grown in the gas phase showed that atomic mixing pattern was pretty stable [55], which is not consistent with our results.

In a study by Darby et al. [28], the Gupta potential was also used for Cu–Au clusters with up to 56 atoms, and results showed that they exhibited primarily icosahedral motifs. The conclusion is consistent with our simulation. Furthermore, the $Cu_{core}Au_{shell}$ segregation tendency was also found by Cheng et al. [56] with the Gupta potential and Monte Carlo method. Wilson et al. [27] performed searching the lowest energy homotops for icosahedral and cuboctahedral Cu–Au nanoalloys, and results showed that for each composition structures tended to have predominantly Au atoms on the surface and Cu atoms in the core. It was further verified by Tran & Johnston [57] while studying all compositions of $Cu_nAu_{38-n}$ clusters by DFT calculations. Furthermore, in a $Cu_{135}Au_{174}$ core–shell cluster calculated using DFT, most surface sites were occupied by Au atoms [58], which is consistent with our results. Moreover, we note that a better parametrization of the Gupta potential was developed by Goh et al. [59], which showed better agreement with DFT results than in the present study.

# 4. Conclusion

The putative stable structures of Ag–Au and Cu–Au clusters with 1 : 1, 1 : 3 and 3 : 1 compositions in the size range of 60–108 are obtained using adaptive immune optimization algorithm with the constructed inner cores (AIOA-IC) method. The many-body Gupta potential is adopted to describe the interaction in bimetallic clusters. Results show that the dominant motifs of Ag–Au and Cu–Au clusters are decahedron and icosahedron, respectively. However, In $Ag_{3n}Au_n$ clusters, i.e. Ag-rich contents, more icosahedra are found than in $Ag_nAu_n$ and $Ag_nAu_{3n}$ clusters. In $Cu_{3n}Au_n$ clusters, i.e. Cu-rich contents, there exist several decahedral motifs. Furthermore, a special Leary tetrahedral motif appears at $Ag_{49}Au_{49}$, $Ag_{50}Au_{50}$, $Ag_{78}Au_{26}$ and $Cu_{75}Au_{25}$. On the other hand, order parameters and bond numbers are calculated to study the atomic distribution. Results show that in all investigated Cu–Au clusters, Cu atoms occupy the inner shell, and Au atoms scatter on the surface, forming $Cu_{core}Au_{shell}$ configurations. In Ag–Au clusters, most of Ag atoms tend to occupy the outer-shell sites, but partial Ag atoms are located in the inner shell, while Au atoms are interconnected with each other in the middle shell.

Data accessibility. The Cartesian coordinates and energies of the putative stable structures of all the investigated Ag–Au and Cu–Au clusters are provided as the electronic supplementary information accompanying this paper.

Authors' contributions. X.W. defined the research topics. R.D., X.W., Y.X., R.C. and T.L. designed methods and modelling, and analysed the data. R.D. designed the codes for data analysis. R.D., Y.X., R.C. and X.W. interpreted the results and wrote the paper. All the authors gave their final approval for publication.

Competing interests. There are no conflicts of interest to declare.

Funding. This study is supported by Key University Science Research Project of Anhui Province (grant nos. KJ2016A859 and KJ2017A349).

Acknowledgements. We are grateful to Xiaofan Li for help with data collection. The authors thank the anonymous reviewers for their constructive suggestions.

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
