## [Reviewer comments · Royal Society Open Science]

Review History

RSOS-190342.R0 (Original submission)

Review form: Reviewer 1

Is the manuscript scientifically sound in its present form?

Yes

Are the interpretations and conclusions justified by the results?

Yes

Is the language acceptable?

Yes

Is it clear how to access all supporting data?

Yes

Do you have any ethical concerns with this paper?

No

Have you any concerns about statistical analyses in this paper?

No

Recommendation?

Major revision is needed (please make suggestions in comments)

Comments to the Author(s)

Referee report on RSOS-190342

Theoretical study of the structures of bimetallic Ag-Au and Cu-Au clusters up to 108 atoms by R. Du, X. Wu, Y. Xu, R. Chen and T. Liu

The authors have used a variation around genetic algorithms together with a many-body potential to determine putative global minima for Ag-Au and Cu-Au clusters at 1:1, 3:1 and 1:3 compositions, containing up to 108 atoms in total. The structures are discussed in terms of their radial distributions and basic high-symmetry motifs such as icosahedra or tetrahedra.

There have been a plethora of similar computational works in the recent decade (global optimization of bimetallic EAM clusters), usually employing genetic algorithms as well, and I think the authors should emphasize the novelty in their methods and results somewhat better. Until then I will postpone my recommendation for publishing this work.

More precisely, I think the following remarks should be addressed in a revised version of this manuscript.

(i) In the present work, biases are often used, by prescribing high symmetry cores. In the description of the method, and although the authors admit about this bias, it would be useful to the readers already familiar with the basics of genetic algorithms what is new and essential in this 'AIOA-IC' method. For instance, how higher in energy are the putative global minima obtained without the 'IC' bias? Also, how robust are the minima obtained, in terms of the chances of success by repeating the global minimization a (small) number of independent times?

(ii) For Au-Ag clusters at 1:1 composition, the authors systematically find energy minima that are lower than those obtained previously by the Johnston group but it was a bit unclear to me whether these differences are qualitative or quantitative. In particular, they may well result from slightly different implementations of the same Gupta potential, even when the same set of EAM parameters are used. For instance, the use of cut-offs with different values (notably in the Johnston group) could explain why lower energies are found here if no cut-off were used to truncate the interactions.

(iii) The radial distance is used as the main order parameter to discuss the different distributions of unlike atoms in the clusters. In this respect, Figure 6 clearly shows that Ag atoms are outside while Au atoms are inside, IN AVERAGE, the opposite trend being found between Au and Cu. Naive visual inspection confirms these trends, but the authors conclude the paper by writing that 'there is no tendency to form core-shell configurations in the investigated Ag-Au clusters'. This requires some clarification!

Perhaps, if the goal is to show that the atoms are broadly distributed throughout the cluster rather than in a core-shell fashion, the second moment of the radial distribution should be used instead of the first moment reflected in equation (4), namely $\langle r_i^2 \rangle - \langle r_i \rangle^2$.

(iv) Gold and silver easily mix in the bulk, and the same is true to a large extent at the nanoscale. Wet chemistry conditions allow core-shell structures to be made, but they are then more likely to be kinetically controlled rather than thermodynamically. Could the authors discuss the energy

gap between the two lowest-energy isomers that they found, and mention whether the second lowest-energy structure is structurally different from the first, or just an homotop?

(v) Please carefully proofread the english when resubmitting.

Review form: Reviewer 2

Is the manuscript scientifically sound in its present form?

Yes

Are the interpretations and conclusions justified by the results?

Yes

Is the language acceptable?

Yes

Is it clear how to access all supporting data?

Yes

Do you have any ethical concerns with this paper?

No

Have you any concerns about statistical analyses in this paper?

No

Recommendation?

Major revision is needed (please make suggestions in comments)

Comments to the Author(s)

This paper demonstrates the nice efficiency of the optimization algorithm proposed by the authors. As listed in Table 2, a lot of new global minima are found. The finding of new global minima is important and worth reporting in the journal. The calculations seem sound and well performed up to my knowledge, since parameters (F. Cleri and V. Rosato) used in the calculations seem adequate for reliable results within the framework of the semi-empirical Gupta calculation. Three different stoichiometries 1:1, 1.3 and 3:1 are considered for Ag-Au and Cu-Au clusters. Furthermore, the difference between geometrical structures and atomic distribution for Ag-Au and Cu-Au clusters is discussed. In summary I recommend publication after revision as explained below.

- 1) For Ag-Au clusters, more results with theoretical calculations should be mentioned.
- 2) This work has carried on the theory research to Ag-Au and Cu-Au clusters. It is suggested that their application prospects should be prospected and even their performance calculation results in fields such as catalysis should be provided.
- 3) The authors should make an in-depth study of the performance advantages and the mechanism of action of bimetallic compounds. Some related papers may be referenced, such as, Chem. Mater., 2014, 26, 3418-3426; J. Mater. Chem. A, 2015, 3, 20973-20982; Nano Res. 2017, 10, 3726-3742; ACS Appl. Mater. Interfaces 2017, 9, 40655-40670; Nanoscale, 2019, Advance Article, DOI: 10.1039/C9NR00962K.
- 4) There are many typo and grammar mistakes. And several references are repeated. Please revise the manuscript carefully.

Decision letter (RSOS-190342.R0)

29-Mar-2019

Dear Professor Wu:

Title: Theoretical study of the structures of bimetallic Ag-Au and Cu-Au clusters up to 108 atoms
Manuscript ID: RSOS-190342

The editor assigned to your manuscript has now received comments from reviewers. We would like you to revise your paper in accordance with the referee and Subject Editor suggestions which can be found below (not including confidential reports to the Editor). Please note this decision does not guarantee eventual acceptance.

Please submit your revised paper before 21-Apr-2019. Please note that the revision deadline will expire at 00.00am on this date. If we do not hear from you within this time then it will be assumed that the paper has been withdrawn. In exceptional circumstances, extensions may be possible if agreed with the Editorial Office in advance. We do not allow multiple rounds of revision so we urge you to make every effort to fully address all of the comments at this stage. If deemed necessary by the Editors, your manuscript will be sent back to one or more of the original reviewers for assessment. If the original reviewers are not available we may invite new reviewers.

Please also include the following statements alongside the other end statements. As we cannot publish your manuscript without these end statements included, if you feel that a given heading is not relevant to your paper, please nevertheless include the heading and explicitly state that it is not relevant to your work.

- Ethics statement

Please clarify whether you received ethical approval from a local ethics committee to carry out your study. If so please include details of this, including the name of the committee that gave consent in a Research Ethics section after your main text. Please also clarify whether you received informed consent for the participants to participate in the study and state this in your Research Ethics section.

OR

Please clarify whether you obtained the necessary licences and approvals from your institutional animal ethics committee before conducting your research. Please provide details of these licences and approvals in an Animal Ethics section after your main text.

OR

Please clarify whether you obtained the appropriate permissions and licences to conduct the fieldwork detailed in your study. Please provide details of these in your methods section.

- Acknowledgements

On behalf of the Subject Editor Professor Anthony Stace and the Associate Editor Professor Kim Jelfs.

RSC Associate Editor:
Comments to the Author:
(There are no comments.)

RSC Subject Editor:
Comments to the Author:
(There are no comments.)

Reviewers' Comments to Author:
Reviewer: 1

Comments to the Author(s)
Referee report on RSOS-190342
Theoretical study of the structures of bimetallic Ag-Au and Cu-Au clusters up to 108 atoms by R. Du, X. Wu, Y. Xu, R. Chen and T. Liu

The authors have used a variation around genetic algorithms together with a many-body potential to determine putative global minima for Ag-Au and Cu-Au clusters at 1:1, 3:1 and 1:3 compositions, containing up to 108 atoms in total. The structures are discussed in terms of their radial distributions and basic high-symmetry motifs such as icosahedra or tetrahedra.

There have been a plethora of similar computational works in the recent decade (global optimization of bimetallic EAM clusters), usually employing genetic algorithms as well, and I think the authors should emphasize the novelty in their methods and results somewhat better. Until then I will postpone my recommendation for publishing this work.

More precisely, I think the following remarks should be addressed in a revised version of this manuscript.

(i) In the present work, biases are often used, by prescribing high symmetry cores. In the description of the method, and although the authors admit about this bias, it would be useful to the readers already familiar with the basics of genetic algorithms what is new and essential in this 'AIOA-IC' method. For instance, how higher in energy are the putative global minima obtained without the 'IC' bias? Also, how robust are the minima obtained, in terms of the chances of success by repeating the global minimization a (small) number of independent times?

(ii) For Au-Ag clusters at 1:1 composition, the authors systematically find energy minima that are lower than those obtained previously by the Johnston group but it was a bit unclear to me whether these differences are qualitative or quantitative. In particular, they may well result from slightly different implementations of the same Gupta potential, even when the same set of EAM parameters are used. For instance, the use of cut-offs with different values (notably in the Johnston group) could explain why lower energies are found here if no cut-off were used to truncate the interactions.

(iii) The radial distance is used as the main order parameter to discuss the different distributions of unlike atoms in the clusters. In this respect, Figure 6 clearly shows that Ag atoms are outside while Au atoms are inside, IN AVERAGE, the opposite trend being found between Au and Cu. Naive visual inspection confirms these trends, but the authors conclude the paper by writing that 'there is no tendency to form core-shell configurations in the investigated Ag-Au clusters'. This requires some clarification!

Perhaps, if the goal is to show that the atoms are broadly distributed throughout the cluster rather than in a core-shell fashion, the second moment of the radial distribution should be used instead of the first moment reflected in equation (4), namely $\langle r_i^2 \rangle - \langle r_i \rangle^2$.

(iv) Gold and silver easily mix in the bulk, and the same is true to a large extent at the nanoscale. Wet chemistry conditions allow core-shell structures to be made, but they are then more likely to be kinetically controlled rather than thermodynamically. Could the authors discuss the energy gap between the two lowest-energy isomers that they found, and mention whether the second lowest-energy structure is structurally different from the first, or just an homotop?

(v) Please carefully proofread the english when resubmitting.

Reviewer: 2

Comments to the Author(s)

This paper demonstrates the nice efficiency of the optimization algorithm proposed by the authors. As listed in Table 2, a lot of new global minima are found. The finding of new global minima is important and worth reporting in the journal. The calculations seem sound and well performed up to my knowledge, since parameters (F. Cleri and V. Rosato) used in the calculations seem adequate for reliable results within the framework of the semi-empirical Gupta calculation. Three different stoichiometries 1:1, 1.3 and 3:1 are considered for Ag-Au and Cu-Au clusters. Furthermore, the difference between geometrical structures and atomic distribution for Ag-Au and Cu-Au clusters is discussed. In summary I recommend publication after revision as explained below.

1) For Ag-Au clusters, more results with theoretical calculations should be mentioned.

2) This work has carried on the theory research to Ag-Au and Cu-Au clusters. It is suggested that their application prospects should be prospected and even their performance calculation results in fields such as catalysis should be provided.

3) The authors should make an in-depth study of the performance advantages and the mechanism of action of bimetallic compounds. Some related papers may be referenced, such as, Chem. Mater., 2014, 26, 3418-3426; J. Mater. Chem. A, 2015, 3, 20973-20982; Nano Res. 2017, 10, 3726-3742; ACS Appl. Mater. Interfaces 2017, 9, 40655-40670; Nanoscale, 2019, Advance Article, DOI: 10.1039/C9NR00962K.

4) There are many typo and grammar mistakes. And several references are repeated. Please revise the manuscript carefully.

Author's Response to Decision Letter for (RSOS-190342.R0)

See Appendix A.

RSOS-190342.R1 (Revision)

Review form: Reviewer 1

Is the manuscript scientifically sound in its present form?

Yes

Are the interpretations and conclusions justified by the results?

Yes

Is the language acceptable?

Yes

Is it clear how to access all supporting data?

No

Do you have any ethical concerns with this paper?

No

Have you any concerns about statistical analyses in this paper?

Yes

Recommendation?

Accept with minor revision (please list in comments)

Comments to the Author(s)

Second report on RSOS190342

Theoretical study of the structures of bimetallic Ag-Au and Cu-Au clusters up to 108 atoms by R. Du, S. Tang, X. Wu, Y. Xu, R. Chen, T. Liu

The authors have taken most of the comments into account, except for some points raised and not

addressed at all. I am listing below some remaining issues that need to be addressed before publication can be recommended.

-One question raised was about statistics: how is the robustness of the method in locating the putative global minima? I was asking about the probability of success by repeating the optimization a number of (independent) times, but got no reply, and the revision shows no change in this respect. The superiority of the algorithm requires such data to be provided for a meaningful evaluation.

-The idea of seeding the geometries with cores with prescribed symmetry is quite old, Hartke already used it more than 20 years ago in a paper on LJ clusters. He then called them 'niches'.

-The order parameter R is introduced to 'explain the atomic distribution or mixing degree of different elements', but I found this paragraph really descriptive rather than providing any explanation. The authors really need to dig deeper.

On a side comment, in their response to earlier criticism, a confusion appears about the many-body potential used in this study (and earlier by Johnston et al.), the 'cut-off' function I was referring to being a truncation of the pair terms for repulsion and in function mimicking the electronic density to some nearest-neighbors. Usually this cut-off proceeds through a continuous function for optimisation purposes, in no way it is related to cut-offs in the DFT calculations, where it refers there to truncation in the amount of plane waves that are used in the basis set.

Review form: Reviewer 2

Is the manuscript scientifically sound in its present form?

Yes

Are the interpretations and conclusions justified by the results?

Yes

Is the language acceptable?

Yes

Is it clear how to access all supporting data?

Yes

Do you have any ethical concerns with this paper?

No

Have you any concerns about statistical analyses in this paper?

No

Recommendation?

Accept as is

Comments to the Author(s)

The author seriously replied to the questions raised and suggested receiving this manuscript.

Decision letter (RSOS-190342.R1)

22-May-2019

Dear Professor Wu:

Title: Theoretical study of the structures of bimetallic Ag-Au and Cu-Au clusters up to 108 atoms
Manuscript ID: RSOS-190342.R1

Thank you for submitting the above manuscript to Royal Society Open Science. On behalf of the Editors and the Royal Society of Chemistry, I am pleased to inform you that your manuscript will be accepted for publication in Royal Society Open Science subject to minor revision in accordance with the referee suggestions. Please find the reviewers' comments at the end of this email.

The reviewers and handling editors have recommended publication, but also suggest some minor revisions to your manuscript. Therefore, I invite you to respond to the comments and revise your manuscript.

Because the schedule for publication is very tight, it is a condition of publication that you submit the revised version of your manuscript before 31-May-2019. Please note that the revision deadline will expire at 00.00am on this date. If you do not think you will be able to meet this date please let me know immediately.

Supplementary files will be published alongside the paper on the journal website and posted on

the online figshare repository (<https://figshare.com>). The heading and legend provided for each supplementary file during the submission process will be used to create the figshare page, so please ensure these are accurate and informative so that your files can be found in searches. Files on figshare will be made available approximately one week before the accompanying article so that the supplementary material can be attributed a unique DOI.

Best wishes,

Dr Laura Smith
Publishing Editor, Journals

On behalf of the Subject Editor Professor Anthony Stace and the Associate Editor Professor Kim Jelfs.

RSC Associate Editor:

Comments to the Author:

The authors need to address the additional points from reviewer 2 that that reviewer is still not satisfied have been satisfactorily answered. If they do this, the paper can be accepted.

RSC Subject Editor:

Comments to the Author:

(There are no comments.)

Reviewer comments to Author:

Reviewer: 2

Comments to the Author(s)

The author seriously replied to the questions raised and suggested receiving this manuscript.

Reviewer: 1

Comments to the Author(s)

Second report on RSOS190342

Theoretical study of the structures of bimetallic Ag-Au and Cu-Au clusters up to 108 atoms by R. Du, S. Tang, X. Wu, Y. Xu, R. Chen, T. Liu

The authors have taken most of the comments into account, except for some points raised and not

addressed at all. I am listing below some remaining issues that need to be addressed before publication can be recommended.

-One question raised was about statistics: how is the robustness of the method in locating the putative global minima? I was asking about the probability of success by repeating the optimization a number of (independent) times, but got no reply, and the revision shows no change in this respect. The superiority of the algorithm requires such data to be provided for a meaningful evaluation.

-The idea of seeding the geometries with cores with prescribed symmetry is quite old, Hartke already used it more than 20 year ago in a paper on LJ clusters. He then called them 'niches'.

-The order parameter R is introduced to 'explain the atomic distribution or mixing degree of different elements', but I found this paragraph really descriptive rather than providing any explanation. The authors really need to dig deeper.

On a side comment, in their response to earlier criticism, a confusion appears about the many-body potential used in this study (and earlier by Johnston et al.), the 'cut-off' function I was referring to being a truncation of the pair terms for repulsion and in function mimicking the electronic density to some nearest-neighbors. Usually this cut-off proceeds through a continuous function for optimisation purposes, in no way it is related to cut-offs in the DFT calculations, where it refers there to truncation in the amount of plane waves that are used in the basis set.

Author's Response to Decision Letter for (RSOS-190342.R1)

See Appendix B.

RSOS-190342.R2 (Revision)

Review form: Reviewer 1

Is the manuscript scientifically sound in its present form?

Yes

Are the interpretations and conclusions justified by the results?

Yes

Is the language acceptable?

Yes

Is it clear how to access all supporting data?

Yes

Do you have any ethical concerns with this paper?

No

Have you any concerns about statistical analyses in this paper?

No

Recommendation?

Accept as is

Comments to the Author(s)

Report on RSOS-190342R2

Theoretical study of the structures of bimetallic Ag-Au and Cu-Au clusters up to 108 atoms by R. Du, X. Wu, Y. Xu, R. Chen and T. Liu

Overall the authors have addressed most of the concerns I raised in my previous report. I still believe some of the discussion is pretty shallow, notably in discussing the radial repartition of atoms through the radial average distance. Perhaps a single indicator combining the radial distance of the two types of atoms would have done the job of telling whether the structures are mixed or of any of the two possible core-shell types.

I now recommend this article for publication in Royal Society Open Science.

Review form: Reviewer 2

Is the manuscript scientifically sound in its present form?

Yes

Are the interpretations and conclusions justified by the results?

Yes

Is the language acceptable?

Yes

Is it clear how to access all supporting data?

Yes

Do you have any ethical concerns with this paper?

No

Have you any concerns about statistical analyses in this paper?

No

Recommendation?

Accept with minor revision (please list in comments)

Comments to the Author(s)

This paper demonstrates the nice efficiency of the optimization algorithm proposed by the authors. As listed in Table 2, a lot of new global minima are found. The finding of new global minima is important and worth reporting in the journal. The calculations seem sound and well performed up to my knowledge, since parameters (F. Cleri and V. Rosato) used in the calculations seem adequate for reliable results within the framework of the semi-empirical Gupta calculation. Three different stoichiometries 1:1, 1.3 and 3:1 are considered for Ag-Au and Cu-Au clusters. Furthermore, the difference between geometrical structures and atomic distribution for Ag-Au and Cu-Au clusters is discussed. In summary I recommend publication after revision as

explained below.

- 1) For Ag-Au clusters, more results with theoretical calculations should be mentioned.
- 2) This work has carried on the theory research to Ag-Au and Cu-Au clusters. It is suggested that their application prospects should be prospected and even their performance calculation results in fields such as catalysis should be provided.
- 3) The authors should make an in-depth study of the performance advantages and the mechanism of action of bimetallic compounds. Some related papers may be referenced, such as, Chem. Mater., 2014, 26, 3418-3426; J. Mater. Chem. A, 2015, 3, 20973-20982; Nano Res. 2017, 10, 3726-3742; ACS Appl. Mater. Interfaces 2017, 9, 40655-40670; Nanoscale, 2019, 11, 6243-6253.
- 4) There are many typo and grammar mistakes. And several references are repeated. Please revise the manuscript carefully.

Decision letter (RSOS-190342.R2)

04-Jul-2019

Dear Professor Wu:

Title: Theoretical study of the structures of bimetallic Ag-Au and Cu-Au clusters up to 108 atoms
Manuscript ID: RSOS-190342.R2

Thank you for submitting the above manuscript to Royal Society Open Science. On behalf of the Editors and the Royal Society of Chemistry, I am pleased to inform you that your manuscript will be accepted for publication in Royal Society Open Science subject to minor revision in accordance with the referee suggestions. Please find the reviewers' comments at the end of this email.

The reviewers and handling editors have recommended publication, but also suggest some minor revisions to your manuscript. Therefore, I invite you to respond to the comments and revise your manuscript. We do not require you to respond to the comments of Reviewer 2.

Please also include the following statements alongside the other end statements. As we cannot publish your manuscript without these end statements included, if you feel that a given heading is not relevant to your paper, please nevertheless include the heading and explicitly state that it is not relevant to your work. We have included a screenshot example of the end statements for reference.

- Acknowledgements

Because the schedule for publication is very tight, it is a condition of publication that you submit the revised version of your manuscript before 13-Jul-2019. Please note that the revision deadline will expire at 00.00am on this date. If you do not think you will be able to meet this date please let me know immediately.

Best wishes,
Dr Laura Smith
Publishing Editor, Journals

On behalf of the Subject Editor Professor Anthony Stace and the Associate Editor Professor Kim Jelfs.

RSC Associate Editor:
Comments to the Author:

(There are no comments.)

RSC Subject Editor:

Comments to the Author:

(There are no comments.)

Reviewer comments to Author:

Reviewer: 2

Comments to the Author(s)

This paper demonstrates the nice efficiency of the optimization algorithm proposed by the authors. As listed in Table 2, a lot of new global minima are found. The finding of new global minima is important and worth reporting in the journal. The calculations seem sound and well performed up to my knowledge, since parameters (F. Cleri and V. Rosato) used in the calculations seem adequate for reliable results within the framework of the semi-empirical Gupta calculation. Three different stoichiometries 1:1, 1.3 and 3:1 are considered for Ag-Au and Cu-Au clusters. Furthermore, the difference between geometrical structures and atomic distribution for Ag-Au and Cu-Au clusters is discussed. In summary I recommend publication after revision as explained below.

- 1) For Ag-Au clusters, more results with theoretical calculations should be mentioned.
- 2) This work has carried on the theory research to Ag-Au and Cu-Au clusters. It is suggested that their application prospects should be prospected and even their performance calculation results in fields such as catalysis should be provided.
- 3) The authors should make an in-depth study of the performance advantages and the mechanism of action of bimetallic compounds. Some related papers may be referenced, such as, Chem. Mater., 2014, 26, 3418-3426; J. Mater. Chem. A, 2015, 3, 20973-20982; Nano Res. 2017, 10, 3726-3742; ACS Appl. Mater. Interfaces 2017, 9, 40655-40670; Nanoscale, 2019, 11, 6243-6253.
- 4) There are many typo and grammar mistakes. And several references are repeated. Please revise the manuscript carefully.

Reviewer: 1

Comments to the Author(s)

Report on RSOS-190342R2

Theoretical study of the structures of bimetallic Ag-Au and Cu-Au clusters up to 108 atoms by R. Du, X. Wu, Y. Xu, R. Chen and T. Liu

Overall the authors have addressed most of the concerns I raised in my previous report. I still believe some of the discussion is pretty shallow, notably in discussing the radial repartition of atoms through the radial average distance. Perhaps a single indicator combining the radial distance of the two types of atoms would have done the job of telling whether the structures are mixed or of any of the two possible core-shell types.

I now recommend this article for publication in Royal Society Open Science.

Author's Response to Decision Letter for (RSOS-190342.R2)

See Appendix C.

Decision letter (RSOS-190342.R3)

16-Jul-2019

Dear Professor Wu:

Title: Theoretical study of the structures of bimetallic Ag-Au and Cu-Au clusters up to 108 atoms
Manuscript ID: RSOS-190342.R3

It is a pleasure to accept your manuscript in its current form for publication in Royal Society Open Science. The chemistry content of Royal Society Open Science is published in collaboration with the Royal Society of Chemistry.

On behalf of the Subject Editor Professor Anthony Stace and the Associate Editor Professor Kim Jelfs.

RSC Associate Editor
Comments to the Author:
(There are no comments.)

Reviewer(s)' Comments to Author:

Appendix A

A list of changes and replies to the reviewer's comments

We first thank the reviewer for his/her positive evaluation and valuable suggestions, and the following is our response to individual problems.

For the reviewer #1's comments:

1. Q: In the present work, biases are often used, by prescribing high symmetry cores. In the description of the method, and although the authors admit about this bias, it would be useful to the readers already familiar with the basics of genetic algorithms what is new and essential in this 'AIOA-IC' method. For instance, how higher in energy are the putative global minima obtained without the 'IC' bias? Also, how robust are the minima obtained, in terms of the chances of success by repeating the global minimization a (small) number of independent times?

A: The reviewer's considerations were reasonable. However, for the structural optimization of Cu-Au and Ag-Au clusters, the efficiency of AIOA-IC method was significantly higher than that of AIOA (without constraint). Therefore, if the constraint is relaxed, under the same run times, the global minimum structures demonstrated in this work would hard to be obtained by AIOA. In fact, in order to enhance the unbiased property of the AIOA-IC method, the starting structures were also randomly generated and local minimized with the same probability, which has been introduced in this manuscript and previous papers (Du RB, Xu YQ, Sun Y, Wu X, Journal of Alloys and Compounds 774 (2019) 677-684, doi: 10.1016/j.jallcom.2018.10.069).

2. Q: For Au-Ag clusters at 1:1 composition, the authors systematically find energy minima that are lower than those obtained previously by the Johnston group but it was a bit unclear to me whether these differences are qualitative or quantitative. In particular, they may well result from slightly different implementations of the same Gupta potential, even when the same set of EAM parameters are used. For instance, the use of cut-offs with different values (notably in the Johnston group)

could explain why lower energies are found here if no cut-off were used to truncate the interactions.

A: According to the comment of the reviewer, we re-examined the potential parameters of the Gupta potential for Ag-Au clusters, and confirmed that the same parameters were taken. The results are quantitatively compared with those by Johnston group, and they are all listed in Table 2. In order to conform that the same parameters were used, we also optimized Ag_nAu_n ($n = 10-30$) clusters with small sizes, and obtained the same structures and energies as in the literature (Chen FY, Johnston RL. 2008 ACS Nano 2, 165-175. doi: 10.1021/nn700226y). Furthermore, the corresponding description was modified from page 7, line -2 to page 8, line 1. On the other hand, we also examined the use of cut-off values, and found that they were mainly used in DFT calculations.

3. **Q:** The radial distance is used as the main order parameter to discuss the different distributions of unlike atoms in the clusters. In this respect, Figure 6 clearly shows that Ag atoms are outside while Au atoms are inside, IN AVERAGE, the opposite trend being found between Au and Cu. Naive visual inspection confirms these trends, but the authors conclude the paper by writing that 'there is no tendency to form core-shell configurations in the investigated Ag-Au clusters'. This requires some clarification! Perhaps, if the goal is to show that the atoms are broadly distributed throughout the cluster rather than in a core-shell fashion, the second moment of the radial distribution should be used instead of the first moment reflected in equation (4), namely $\langle r_i^2 \rangle - \langle r_i \rangle^2$.

A: The reviewer was right. The sentence of "Therefore, there is no tendency to form core-shell configurations in the investigated Ag-Au clusters." was inaccurate, and it was deleted in the revised manuscript.

4. **Q:** Gold and silver easily mix in the bulk, and the same is true to a large extent at the nanoscale. Wet chemistry conditions allow core-shell structures to be made, but they are then more likely to be kinetically controlled rather than

thermodynamically. Could the authors discuss the energy gap between the two lowest-energy isomers that they found, and mention whether the second lowest-energy structure is structurally different from the first, or just an homotop?

A: For the first lowest-energy isomer and the second lowest-energy structure of Ag_nAu_n clusters, there indeed exist structural differences in some cases. For example, as plotted below, in our optimized $\text{Ag}_{30}\text{Au}_{30}$ cluster, the energy of the first lowest-energy structure was -188.3738 eV, and the corresponding one of the second lowest-energy structure was -188.3569 eV. The energy difference was only 0.0169 eV, but the structures were very different.

$\text{Ag}_{30}\text{Au}_{30}$

Global minimum
 E : -188.3738 eV

Local minimum
 E : -188.3569 eV

5. **Q:** Please carefully proofread the english when resubmitting.

A: According to the suggestion of the reviewer, English language was modified through the manuscript.

For the reviewer #2's comments:

1. **Q:** For Ag-Au clusters, more results with theoretical calculations should be mentioned.

A: According to the suggestion of the reviewer, more references on Ag-Au clusters were added, and the corresponding description was modified on page 3, lines 11-13.

2. **Q:** This work has carried on the theory research to Ag-Au and Cu-Au clusters. It is

suggested that their application prospects should be prospected and even their performance calculation results in fields such as catalysis should be provided.

A: The Ag-Au and Cu-Au clusters have been widely studied in the fields of medicine, biology, catalysis, optics, and electronics as introduced in our manuscript. The reviewer suggested a valuable question for the application of our results in fields such as catalysis. But, it was indeed hard for us to explore this problem.

- 3. Q:** The authors should make an in-depth study of the performance advantages and the mechanism of action of bimetallic compounds. Some related papers may be referenced, such as, *Chem. Mater.*, 2014, 26, 3418-3426; *J. Mater. Chem. A*, 2015, 3, 20973-20982; *Nano Res.* 2017, 10, 3726-3742; *ACS Appl. Mater. Interfaces* 2017, 9, 40655-40670; *Nanoscale*, 2019, DOI: 10.1039/C9NR00962K.

A: According to the comment of the reviewer, some papers on the bimetallic compounds were referenced. Since bimetallic compounds were not covered in this manuscript, relevant descriptions were modified on page 2, lines 4-5, and 7.

- 4. Q:** There are many typo and grammar mistakes. And several references are repeated. Please revise the manuscript carefully.

A: According to the suggestion of the reviewer, English language was modified through the manuscript, and the references were examined.

Appendix B

A list of changes and replies to the reviewer's comments

We first thank the reviewer for his/her positive evaluation and valuable suggestions, and the following is our response to individual problems.

For the reviewer #1's comments:

1. Q: One question raised was about statistics: how is the robustness of the method in locating the putative global minima? I was asking about the probability of success by repeating the optimization a number of (independent) times, but got no reply, and the revision shows no change in this respect. The superiority of the algorithm requires such data to be provided for a meaningful evaluation.

A: According to the suggestion of the reviewer, the number of runs (over 100 independent runs) for locating the stable structures of Ag-Au and Cu-Au clusters up to 108 atoms were added. Considering that the description of structural optimization for the investigated clusters was introduced in different paragraphs of the manuscript, we added the corresponding successful rates to the supplementary files. On the other hand, a brief description of successful rates for the algorithm was added on page 11, lines 1-5.

2. Q: The idea of seeding the geometries with cores with prescribed symmetry is quite old, Hartke already used it more than 20 year ago in a paper on LJ clusters. He then called them 'niches'.

A: The reviewer was right, and this reminder was helpful. The corresponding literature was cited and described on page 6, line -1.

3. Q: The order parameter R is introduced to 'explain the atomic distribution or mixing degree of different elements', but I found this paragraph really descriptive rather than providing any explanation. The authors really need to dig deeper.

A: We thank the reviewer for his/her comments. But we have not found a better way to describe the atomic distribution. The order parameter R described in this

manuscript was also used in the reported works, such as “N. T. Wilson, R. L. Johnston, A theoretical study of atom ordering in copper-gold nanoalloy clusters. *J. Mater. Chem.*, 2002, 12, 2913-2922, doi: 10.1039/b204069g”.

4. **Q:** On a side comment, in their response to earlier criticism, a confusion appears about the many-body potential used in this study (and earlier by Johnston et al.), the 'cut-off' function I was referring to being a truncation of the pair terms for repulsion and in function mimicking the electronic density to some nearest-neighbors. Usually this cut-off proceeds through a continuous function for optimization purposes, in no way it is related to cut-offs in the DFT calculations, where it refers there to truncation in the amount of plane waves that are used in the basis set.

A: We consulted the related literatures (such as N.T. Wilson, M.S. Bailey, R. L. Johnston, Atom ordering in cuboctahedral Ni-Al nanoalloys. *Inorganica Chimica Acta*, 2006, 359, 3649-3658, doi: 10.1016/j.ica.2006.02.029) and concluded that the reviewer was right. However, we did not use cut-off constraint in the Gupta potential. The cut-off values indeed reflected pairwise repulsive term and the attractive many-body terms. Apparently, if the cut-off values were used in the optimization of the stable structures of Ag-Au clusters of 1:1 composition up to 150 atoms by Johnston, the obtained results might be different from the corresponding ones without cut-off constraint. Therefore, according to the comment of the reviewer, the additional description was added on page 8, lines 12-16.

Appendix C

A list of changes and replies to the reviewer's comments

We first thank the reviewer for his/her positive evaluation and valuable suggestions, and the following is our response to individual problems.

For the editor's comments:

- 1. Q:** Please also include the following statements alongside the other end statements. As we cannot publish your manuscript without these end statements included, if you feel that a given heading is not relevant to your paper, please nevertheless include the heading and explicitly state that it is not relevant to your work. We have included a screenshot example of the end statements for reference.

Acknowledgements.

A: According to the suggestion, the declaration of Acknowledgements was added in the revised manuscript.

For the reviewer #1's comments:

- 1. Q:** Overall the authors have addressed most of the concerns I raised in my previous report. I still believe some of the discussion is pretty shallow, notably in discussing the radial repartition of atoms through the radial average distance. Perhaps a single indicator combining the radial distance of the two types of atoms would have done the job of telling whether the structures are mixed or of any of the two possible core-shell types.

A: After consulting the literature and thinking carefully, we could not find a more appropriate way to describe the segregation pattern. Furthermore, the core-shell pattern in Ag-Au and Cu-Au clusters could indeed be seen through the radial average distance. So, no more descriptions would be added.